# A Study on the Influence of Double Tunnel Excavations on the Settlement Deformation of Flood Control Dikes

Qingxiong Zhao [1], Kaihui Li [1,*], Ping Cao [1], Yinzhu Liu [1,2], Yongkang Pang [1] and Jingshuo Liu [3,*]

[1] School of Resources and Safety Engineering, Central South University, Changsha 410083, China
[2] Dexing Copper Mine, Jiangxi Copper Co., Ltd., Dexing 334224, China
[3] Hunan Polytechnic of Water Resources and Electric Power, Changsha 410131, China
* Correspondence: geokhli@csu.edu.cn (K.L.); ljs7327@163.com (J.L.)

**Abstract:** This article combines numerical simulation and field monitoring methods to study the stability of the overlying Liuyang River embankment in the tunnel crossing between Huaqiao Station and Rice Museum Station of Changsha Metro's Line 6. Using AutoCAD, 3Dmine, and COMSOL Multiphysics, a calculation model of the entire subway tunnel section crossing the flood control embankment under the coupling of fluids and solids was established. The process of tunnel-crossing the embankment and the variation in spatial displacement and plastic strain in different geological layers were analyzed from the perspective of time evolution and spatial distribution. The research results show that during the process of crossing the embankment, the deformation of the east bank is greater than that of the west bank, and crossing the west bank is the relatively riskier stage of the entire project. Moreover, during the process of crossing the embankment, the overlying soil layer will produce a plastic strain zone, and only a small amount of plastic strain is generated in the surrounding sandstone layer of the tunnel walls. In terms of the magnitude of plastic strain, the plastic strain area produced by the leading tunnel's surrounding rocks is larger than that of the following tunnel. As the excavation progresses, a funnel-shaped settlement displacement gradually forms during the passage of the leading tunnel, and this settlement funnel gradually expands during the passage of the following tunnel, with the maximum settlement point transitioning from directly above the leading tunnel to the middle position between the two tunnels. Using the jitter filter algorithm and the adjacent average method to process the field monitoring data, the results show that the monitored deformation results well match the simulated settlement results.

**Keywords:** numerical simulation; flood control embankment; river-crossing tunnel; displacement settlement; fluid–solid coupling; plastic deformation

## 1. Introduction

When a city constructs a tunnel underneath a river or a river embankment, the excavation poses a threat to the stability of the embankment, while the stability and safety of buildings around rivers and the sustainability of ecological development are also greatly challenged. In previous construction cases, there have been engineering examples of tunnel construction causing damage and seepage in flood control embankments [1]. For instance, the leakage of a flood control embankment occurred in Nanjing during the construction of a tunnel crossing the Yangtze River, which later required extensive manpower and resources for reinforcement and repair; it affected the sustainable development of the surrounding ecological environment [2]. In Hangzhou, the collapse of the tunnel opening occurred due to significant variations in water pressure and consecutive days of rainfall, resulting in the increased settlement of the embankment [3]. Therefore, it is crucial to analyze the construction process, predict the surface deformation caused by the tunnel's excavation in the early stages, and assess the stability of the tunnel during construction under the coupling of the seepage and stress fields.

Tunnel excavation under a river typically involves the dual problems of rock mechanics and underground water seepage. Yan et al. [4] proposed an analytical solution for the external stress of circular TBM tunnel lining considering a seepage force based on the theories of elasticity and seepage. Shi et al. [5] used COMSOL Multiphysics software to simulate the flow–solid coupling of layered anisotropic rock slopes and compared this method with traditional model methods based on the actual conditions of the Fushun Xiluoti mine slope, showing a better performance in simulating similar problems. Tang et al. [6] analyzed the effects of shallow-buried excavation, a small grouting zone, and initial support lining permeability on the surface settlement in the overlapped sections of the Shenzhen Metro's Line 5 and Line 7 under seepage conditions. The results show that water pressure has a significant influence on the stability of surrounding rocks during tunnel excavation.

Furthermore, due to the high risks posed by tunnel excavation to surface and structural stability, many researchers have used theoretical analysis methods [7–11], field measurements [12–14], laboratory model tests [15–17], and numerical simulations [18–21] to analyze the mechanisms and patterns of ground deformation caused by tunnel excavation. In terms of numerical simulations, Mou et al. [22] studied the lateral surface settlement patterns of double-line shield tunnel construction by analyzing factors such as soil pressure, deformation property of soil, internal friction angles, and tunnel burial depth based on the numerical analysis of a shield tunnel section of the Nanning Metro's Line 3. Yao et al. [23] conducted a study using the three-dimensional finite element method to simulate the excavation process of a shield tunnel near a railway embankment transition zone in Hangzhou, analyzing the effects of tunnel construction on the dynamic stress distribution and differential settlement of adjacent embankment transition zones. They found that due to significant differential settlement caused by tunnel excavation, the embankment needed to be reinforced. He et al. [24] established a three-dimensional analytical model and verified the model through numerical simulations, demonstrating its accuracy by studying the influence of mutual interactions between dual tunnels on ground vibrations. Agbay et al. [25] conducted numerical simulations on different sections along the dual tunnel alignment considering the pre-support system and found that the deformation modulus of the surrounding ground is the main factor determining the ground's settlement. Akbari et al. [26] obtained four numerical models, including single tunnels and double tunnels, using FLAC3D. The study found that the interaction between tunnels is related to the distance between them, with smaller effects at longer distances. Numerically simulating on-site construction conditions is a useful tool for studying the effects of tunnel excavation on the surface and structural stability. However, most studies assume uniform geological strata and lack complex 3D models based on actual conditions. Comprehensive studies on the entire subway section are also lacking. Deng et al. [27] developed a three-dimensional finite element numerical model in a composite stratum using the software "COMSOL". Through the utilization of fluid–structure coupling and multi-physics field analysis, the study derived the laws governing ground settlement deformation and examined the impact of four factors, e.g., tunnel face spacing, tunnel spacing, tunnel depth, and water level, on the riverbed and on the ground settlement curve of double-hole tunnels. Ying et al. [28] used COMSOL to establish a numerical model and carried out parameter analysis on the seabed's thickness, permeability coefficient, tunnel buried depth, and the lining thickness involved in the construction of subsea tunnels. COMSOL mutiphysics simulation results showed that the error caused by the seabed's thickness could not be ignored in tunnel construction.

In this paper, the construction of the Changsha Rail Transit Line 6, which crosses the Liuyang River through a tunnel between Huaqiao Station and the Rice Museum Station, is taken as the background. The three-dimensional geological model and tunnel engineering model of the entire interval tunnel crossing the flood control embankment are established using AutoCAD and 3Dmine. COMSOL Multiphysics software is used for the finite element mesh division and finite element solution, and the curve equation for simulating the actual excavation conditions of the tunnel is established under parameter control. The variation of

the tunnel displacement field throughout the process of crossing the embankment and the Liuyang River is studied. Additionally, the impact law of excavation on the displacement settlement of the top of the flood control embankment is analyzed. The monitoring data from the on-site GPS dynamic long-term monitoring station is compared with the results of numerical simulation.

## 2. Project Overview

The section of the Changsha Metro's Line 6 that crosses the Liuyang River is located between Huaqiao Station and the Rice Museum Station. The segment adopts a parallel double-line shield tunnel construction method to cross the Liuyang River. This construction section passes under the Liuyang River from ZCK40 + 750 to ZCK41 + 000. The shield tunnel starts from Huaqiao Station and moves east along the south side of Renmin Road, passes under the Liuyang River and the flood control embankments on both sides, and reaches the Rice Museum Station via the road along the river. Most of the section runs parallel to the Liuyang River–Guitang River Bridge. The left and right lines of the section are two separated single-line tunnels, as shown in Figure 1. The vertical cross sections of the left and right tunnels are "V"-shaped, with a minimum radius of curvature of 400 m. The distance between the left and right lines is 13.2–15.2 m, and the tunnel excavation diameter is 6.2 m. The tunnel depth ranges from 9.2 m to 25.9 m. The water level of the Liuyang River is between 28.0 m and 29.5 m, with an average elevation of 28.68 m. The water depth ranges from 0.90 m to 5.50 m, with an average depth of 3.64 m. The width of the river is about 220 m. The river water and groundwater are mainly supplied to each other through the cobble layer with close hydraulic connection.

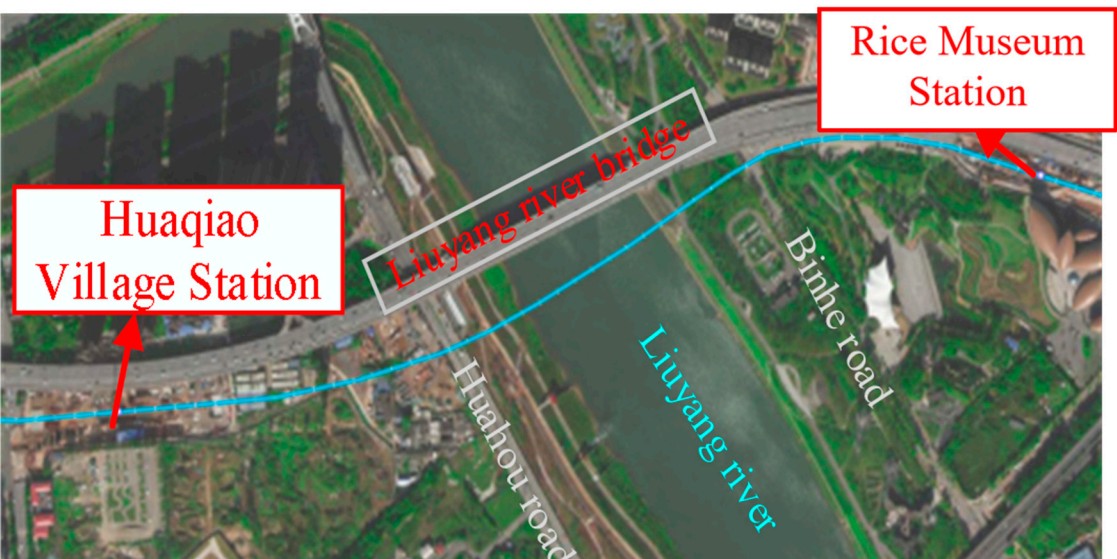

**Figure 1.** Traffic location map of Huaqiao Station–Rice Museum Station of Changsha Rail Transit Line 6 Project.

### 2.1. Engineering Geological Conditions

The tunnel in this section mainly passes through moderately weathered silty fine sandstone, with overlying layers, including a cobble layer, fine sand, silt, silty clay, silted fill, and miscellaneous fill. The silted fill layer has lower strength and is the main part of the embankment. The silt layer has good permeability and acts as an important drainage layer. The cobble layer, with the best permeability, is the main water-conductive layer. The moderately weathered silty fine sandstone layer at the bottom is the main part of the model, and the tunnel construction mainly takes place in this layer, which has relatively good rock integrity and stable surrounding rock. The moderately weathered silty fine sandstone is cohesive and difficult to break by hand. The strength of the sandstone decreases if it is

waterlogged for a long time and cracks can develop if it is exposed for a long period of time. The sandstone becomes soft and disintegrates with alternating wet and dry conditions, thus belonging from extremely soft rock to soft rock. Therefore, when water infiltrates the rock layers, it weakens the surrounding rock, which poses a substantial risk to the stability of the tunnel. This layer is widely distributed horizontally in this section, serving as the stable underlying rock of the site, with an average thickness of 17.97 m.

### 2.2. Hydrogeological Conditions

According to the specific geological structure and rock permeability of the project, tunnel excavation will affect the stress field of the overlying flood control embankment rock and the permeability coefficient of the rock and soil. Meanwhile, the changes in the seepage field will also cause changes in the stress field of the surrounding rock and soil. Accordingly, the stress field and the seepage field are mutually affected, meaning that they are coupled. Changes in the physical field around the rock and soil will affect the displacement of the flood control embankment. The Liuyang River experiences significant water level fluctuations and high flow rates throughout the year, making the hydrogeological conditions complex. Therefore, it is essential to analyze the stability of the flood control embankment of the Liuyang River.

## 3. Model Establishment

(1)    Establishment of Geological Model

The geological model used in numerical simulations has the same dimensions as the actual geological model. Simplifying the computational model is necessary for complex geological conditions. The actual engineering geological conditions are complex and variable, with non-continuous rock and soil layers. Using the actual geological model would increase the computational load while reducing accuracy. The modeling process is shown in Figure 2 and described as follows: (1) Simplify the stratigraphy by retaining the characteristic layers of moderately weathered silty fine sandstone, cobble, silt, and silted fill; (2) Use CAD to convert the plan view into a three-dimensional profile, obtaining the elevation control points of the stratigraphy and filling the missing layers of the borehole profiles; (3) Convert the profiles in 3Dmine software, obtaining a layered geological model by dividing the geological body with the stratigraphic boundaries; (4) Import the three-dimensional geological map into COMSOL software and assign different entity properties to different layers, as shown in Table 1; (5) Mesh and divide the model into a finite number of tetrahedral elements. Due to the limitations of the geological model and computational capacity, an overly detailed model is unnecessary. The final meshing parameters are as follows: a maximum size of 10 m, a minimum size of 0.6 m, a maximum growth rate of adjacent elements of 1.5, a curvature factor of 0.6, and a thin domain resolution of 0.02. The final mesh consists of 510,475 tetrahedral domain elements, 84,114 triangular elements, 8844 boundary elements, and 476 vertex elements;

(2)    Model Setup

To simplify the calculations, each geological layer is treated as homogeneous and isotropic. The failure of moderately weathered silty fine sandstone follows the Hoek–Brown criterion, while the plastic behavior of the cobble, silt, and silted fill layers follows the Mohr–Coulomb criterion. For the flow field, Darcy's law is assumed to hold since the groundwater pressure gradient is not significant. The coupling conditions are assumed to follow the Biot–Willis law;

(3)    Determination of Computational Parameters

Based on preliminary exploration results and laboratory experiments, the mechanical parameters and relevant model calculation parameters for each rock and soil layer were determined. The simulated parameters are listed in Table 1, where the parameters are assigned to their corresponding geological layers;

(4)    Assignment of Initial Conditions

Constructing the initial ground stress field: The model's base is set as a fixed constraint with all side boundaries as roller supports to restrict lateral displacement; apply the force from the water in the riverbed as a boundary load on the riverbed surface, set the water pressure at both ends and the riverbed, and then proceed with the model's equilibrium under the effect of gravity to obtain the initial ground stress field. This is assigned as the initial condition for excavation;

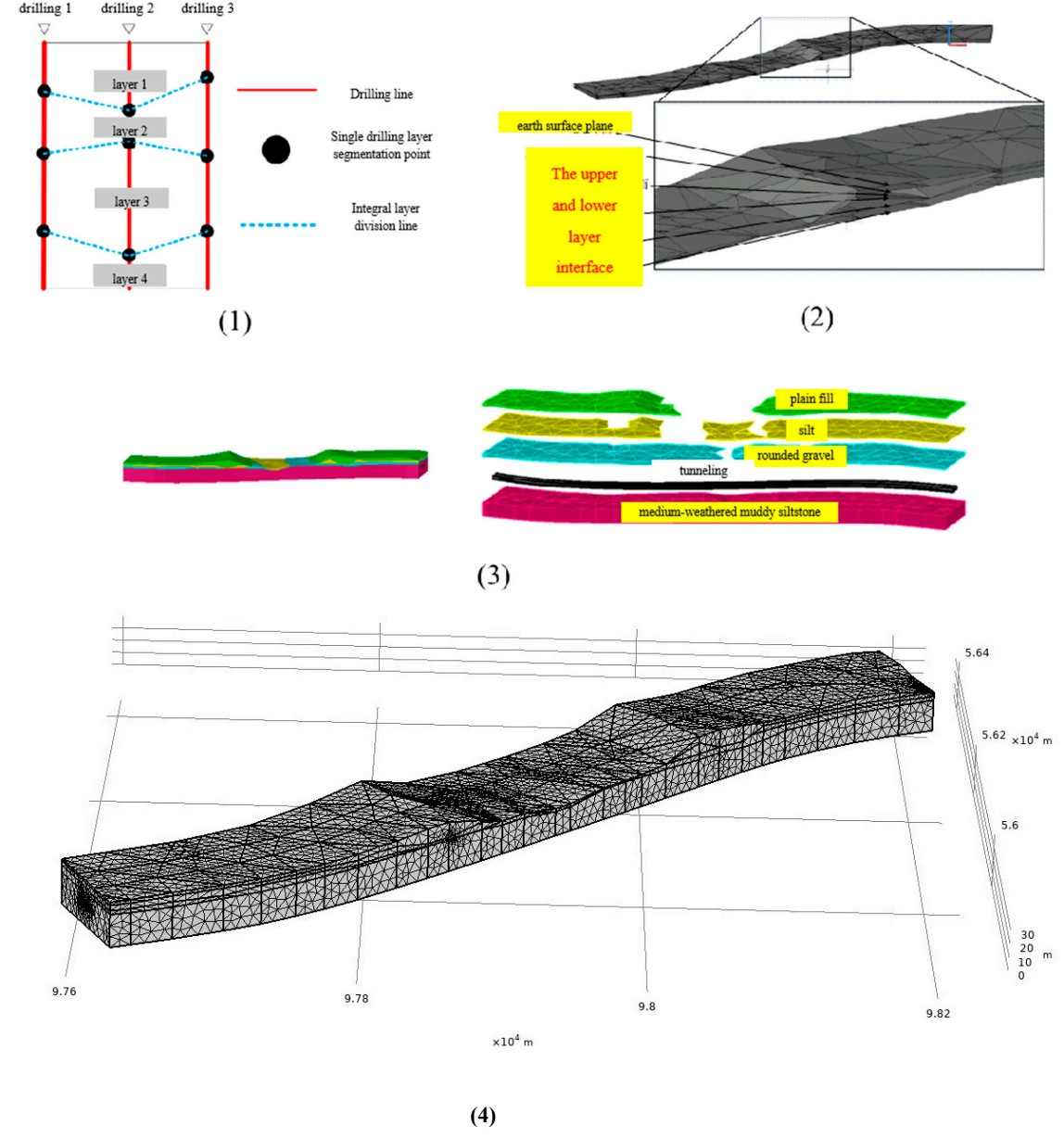

**Figure 2.** Schematic graph of modeling process: (**1**) schematic graph of stratum division; (**2**) result graph of stratum division; (**3**) stratigraphic division in COMSOL; (**4**) result of finite element grid division.

**Table 1.** Simulated parameters.

| Soil/Rock Name | Volume Modulus | Shear Modulus | Density | Porosity | Permeability | Uniaxial Compressive Strength | Hoek–Brown Parameter | | Cohesion | Internal Friction Angle | Biot–Willis Coefficient |
|---|---|---|---|---|---|---|---|---|---|---|---|
| | | | | | | | m | s | | | |
| | MPa | MPa | g/cm$^3$ | | | MPa | | | kPa | ° | |
| Silted fill | 6.72 | 4.23 | 2.01 | 0.09 | $7.09 \times 10^{-14}$ | \ | \ | \ | 24 | 16 | 1 |
| Silt | 6.38 | 3.83 | 2.05 | 0.1 | $9.45 \times 10^{-12}$ | \ | \ | \ | 18 | 23 | 1 |
| Cobble | 33.33 | 25 | 2.2 | 0.3 | $1.12 \times 10^{-11}$ | \ | \ | \ | 2 | 35 | 1 |
| Moderately weathered silty fine sandstone | 393.81 | 415 | 2.45 | 0.23 | $1.77 \times 10^{-13}$ | 21.92 | 10 | 0.357 | \ | \ | 0.85 |

Note: The Biot–Willis coefficient is based on reference values, with a value of 0.85 for the moderately weathered silty fine sandstone (similar to Boise sandstone) and 1 for other soil layers.

(5)    Setting Boundary Conditions

Seepage boundary conditions: Set water pressure boundaries at the west, east, and riverbed locations of the model. According to geological data, the average annual groundwater level on the west side of the Liuyang River is approximately 30 m, while the riverbed water level is around 29 m, and it is 28 m on the east side. After setting the water pressures, the remaining boundaries are set as no flow boundaries;

(6)    Solid mechanics boundary conditions

To obtain the initial ground stress before excavation, the model's bottom is set as a fixed constraint and the surrounding boundaries are set as roller supports. The force from the water in the riverbed is applied as a boundary load on the riverbed surface, and then the entire model is allowed to settle under the effect of gravity to obtain the initial stress state, as shown in Figure 3.

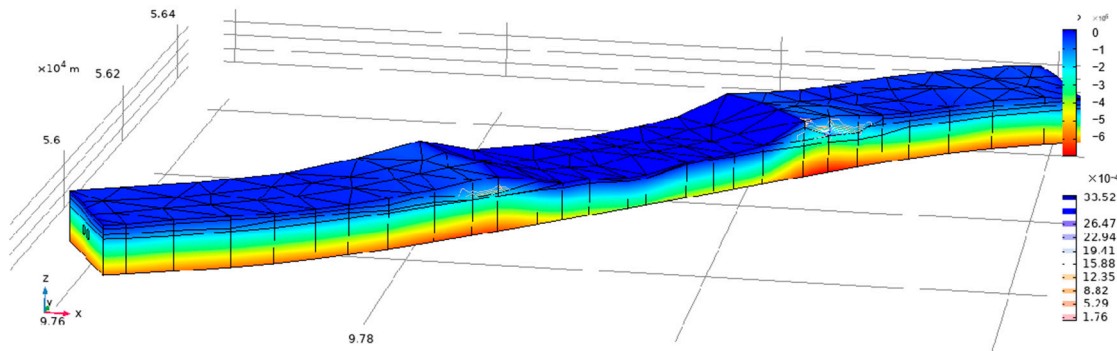

**Figure 3.** Initial ground stress distribution cloud map.

(7)    Setting the Excavation Scenario

The right line is assumed to be 100 m ahead of the left line. The excavation step size is set to 10 m. In COMSOL Multiphysics, excavation can be achieved using the "Activate" module. The idea is to set an activation multiplier for the part of the model that needs to be excavated. When excavation is required, the elastic matrix at the corresponding computational node is multiplied by the activation multiplier, and the density is multiplied by the square of the activation multiplier, effectively removing its load-bearing capacity. To achieve the parameter-controlled excavation, a parameter "depth" is defined as the length of the excavated tunnel. The "origin" is used to obtain the polynomial fit curve of the tunnel, as shown in Figure 4. The curve equations $f(x)$ are as follows:

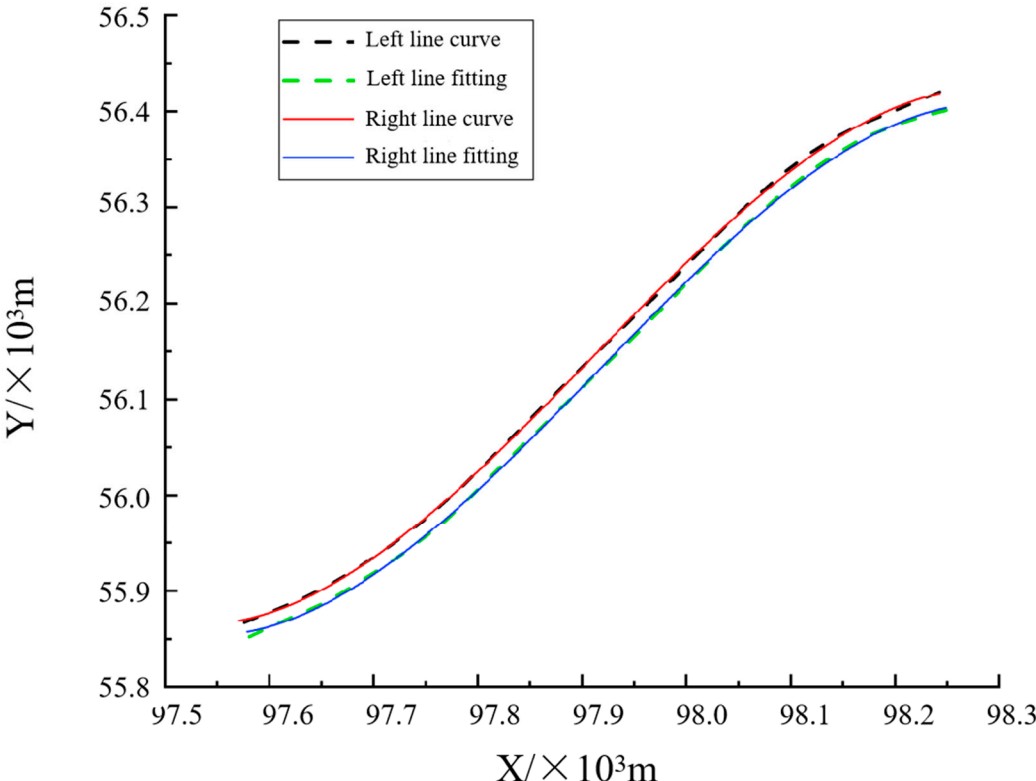

**Figure 4.** Tunnel curve equation fitting.

Left Line:

$$f(x) = 2.47 \times 10^9 - 75536.05x + 0.77x^2 - 2.63 \times 10^{-6}x^3. \tag{1}$$

Right Line:

$$f(x) = 2.51 \times 10^9 - 76787.76x + 0.78x^2 - 2.67 \times 10^{-6}x^3. \tag{2}$$

The X value represents the coordinate value of the tunnel in the east–west direction in the model and corresponds to the X coordinate value of the model. Y(f(x)) represents the coordinate value of the tunnel in the north–south direction in the model.

The integrated curve length is given by the following equation:

$$\text{length} = \int_{x_0}^{x} \sqrt{x^2 + f(x)^2} dx. \tag{3}$$

To determine whether a computational node is activated, the "length" of the computational node is compared with the "depth" parameter. By changing the value of the "depth" parameter, the excavation progress can be controlled.

To enable continuous excavation, the results of the previous excavation step need to be used as the initial condition for the next step. This can be achieved by using the parameterized sweep function. The "depth" parameter is set as a series of parameters. For example, the range of (10, 10, 700) indicates an initial step of excavating 10 m with a step size of 10 m, up to a total length of 700 m. For step-by-step excavation, the prestress of the newly excavated part in each step is set to zero, while the remaining part retains the prestress from the stable stress distribution after the previous excavation step.

## 4. Results and Analysis

### 4.1. Displacement Variation along the Tunnel Alignment during Crossing of the West Bank Embankment

The displacement cloud maps of the west bank embankment at different distances from the tunnel excavation are shown in Figure 5. From the displacement monitoring cloud maps, it can be observed that when the shield machine is not yet close to the flood control embankment, the overlying soil layer is shallow and the pressure within the soil layer is small. Accordingly, the zone of plastic strain within the soil layer is also small, mainly distributed in the cobble layer. When the shield machine approaches and crosses the embankment, the overlying soil layer is thicker compared to the same layer, and the soil pressure is relatively higher. It can be found that the overlying soil layer undergoes a significant plastic strain during the process of crossing the embankment. The tunnel as a whole is located within the moderately weathered silty fine sandstone layer, with only a small amount of plastic strain occurring in the roof and floor of tunnel, while no large-scale plastic strain zone is observed elsewhere. Due to the prior excavation of the tunnel, a significant displacement deformation occurs in the soil layer above the latter tunnel. Furthermore, during the excavation of the latter tunnel, the deformation in the underlying soil layer is larger compared to that in the soil layer above the earlier tunnel. This is mainly attributed to the disturbance caused by the previous tunnel excavation in the soil layer above the latter tunnel, which results in an initial deformation in the soil layer above the latter tunnel. Overall, during the entire process of crossing the embankment, no significant damage occurs.

### 4.2. Displacement Variation along the Tunnel Alignment during Crossing of the East Bank Embankment

The displacement cloud maps of the east bank embankment at different distances from the tunnel excavation are shown in Figure 6. According to the displacement change cloud maps, the deformation patterns of the tunnel sections on the east and west bank embankments under excavation are similar. However, during the excavation of the latter tunnel, the deformation in the underlying soil layers is larger compared to that in the soil layers above the earlier tunnel. The deformation impact on the east bank embankment is relatively more pronounced when compared to the west bank embankment. During the process of crossing the east bank embankment, the overlying soil layer experiences plastic strain zones, and the roof and floor of tunnel undergo small-scale plastic deformation as well.

### 4.3. Final Distribution of Section Displacement

The completed excavation of the tunnel results in displacements along the tunnel section of the left and right lines as shown in Figures 7 and 8. The displacements are particularly significant above the tunnel near the flood control embankments on both sides. The cobble, silt, and silted fill layers within the embankments all show plastic strain zones. However, the moderately weathered silty fine sandstone layer only exhibits minimal plastic strain at the tunnel walls and floor. Additionally, in the distribution of section displacements after tunnel excavation, the displacements are more pronounced near the east bank embankment when compared to the other locations. Correspondingly, the plastic strain is more concentrated within this area. It is attributed to the higher elevation of the east bank embankment and the corresponding greater tunnel depth and overlying soil thickness. As a result, the soil pressure is higher, resulting in larger displacements due to the disturbance caused by the tunnel's excavation. On the other hand, when the tunnel's depth is lower, the arching effect of the soil can mitigate the surface's settlement, leading to smaller deformations [27]. Accordingly, the east bank embankment poses a relatively higher risk in the entire project and should be given special attention during construction.

Total Displacement of Left (trailing) line Section　　　　Total Displacement of right (leading) line Section

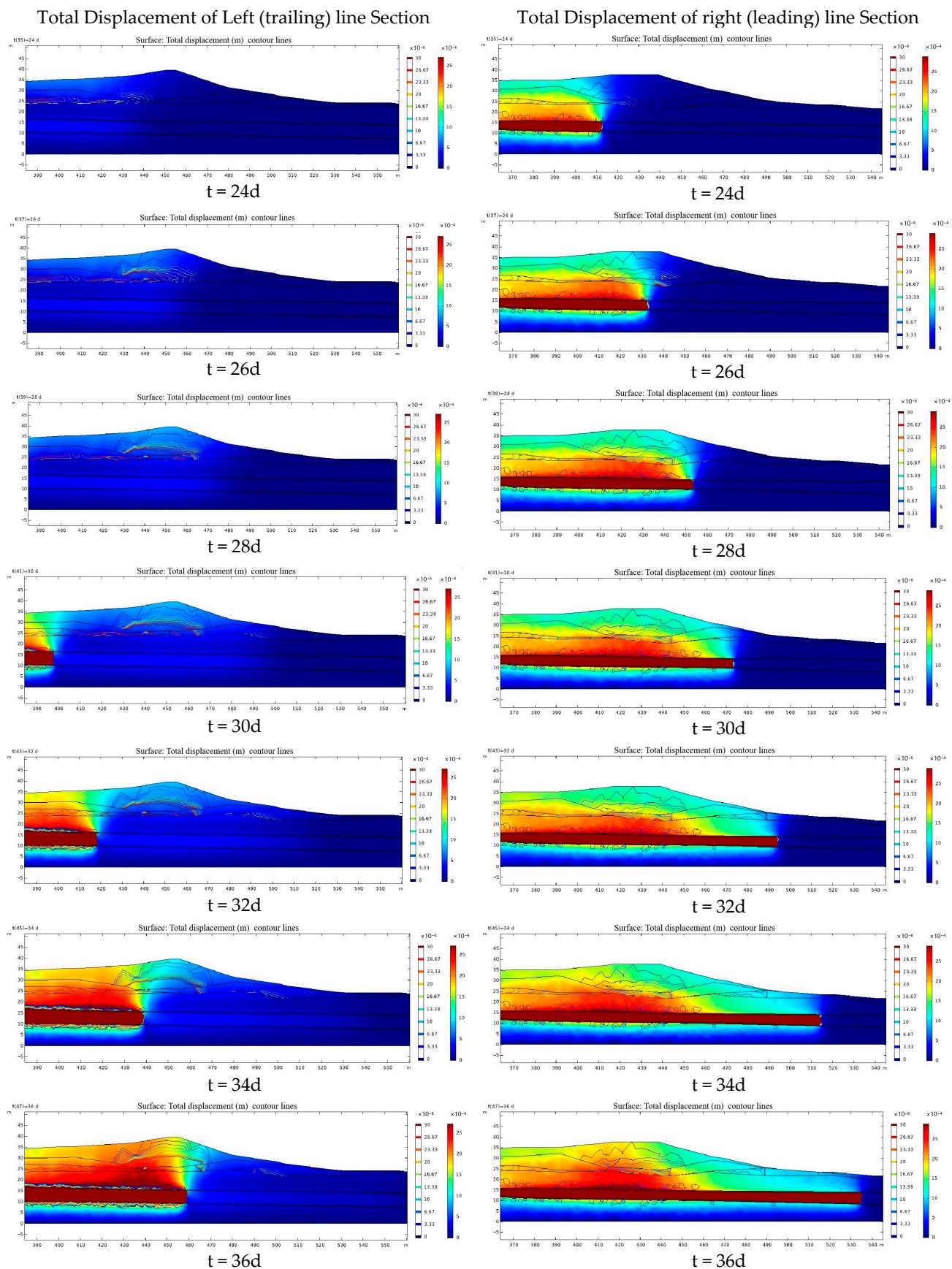

**Figure 5.** Cloud change in longitudinal section of dike tunnel on the west bank.

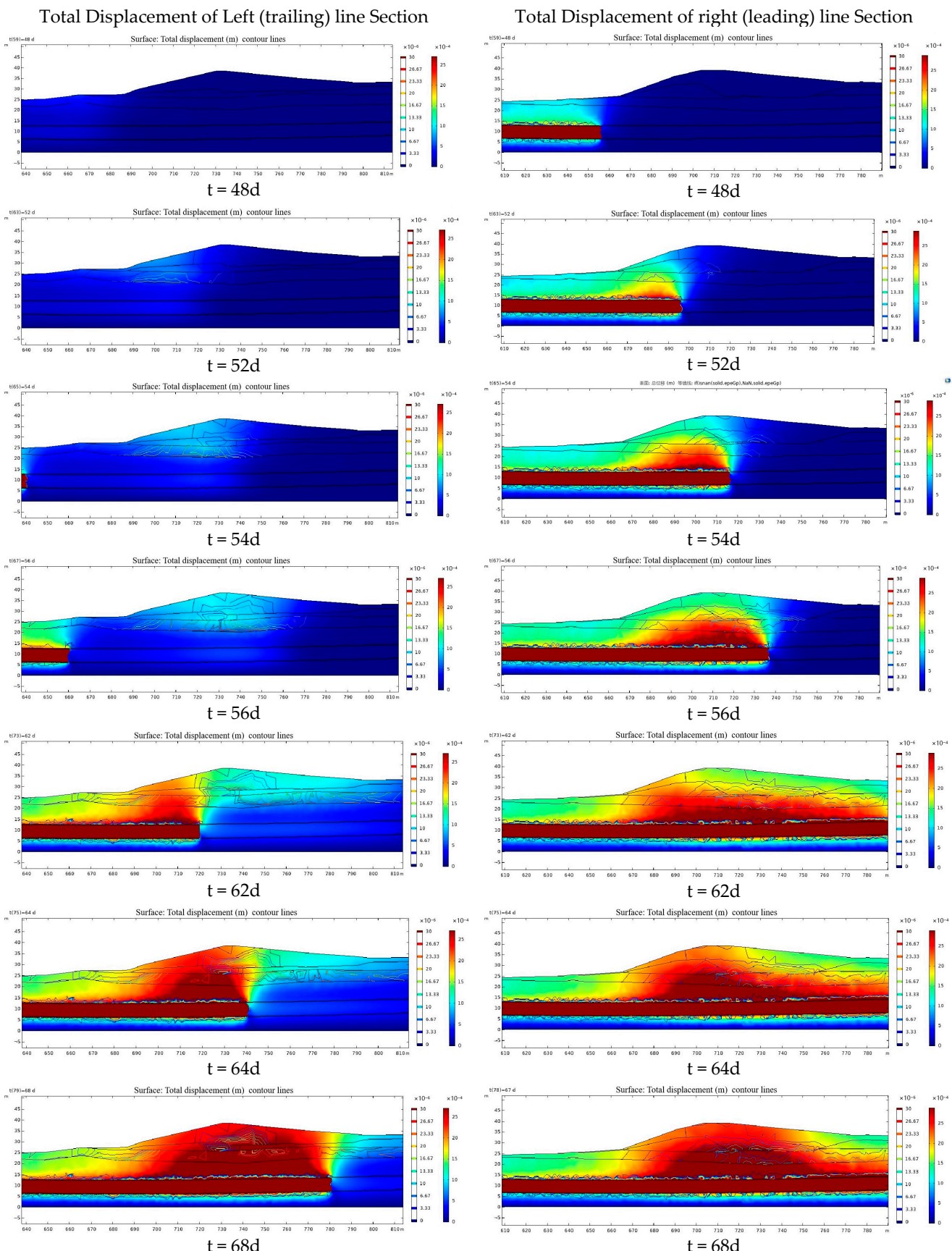

**Figure 6.** Cloud change in longitudinal section of dike tunnel on the east bank.

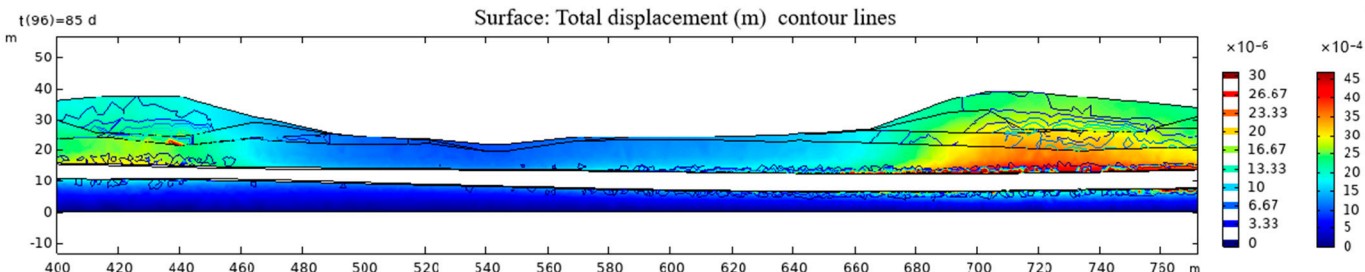

**Figure 7.** Section displacement and plastic strain distribution of the right line.

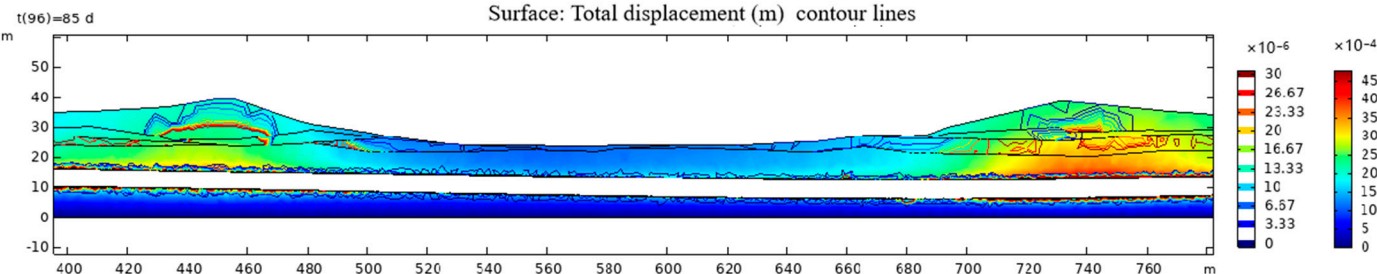

**Figure 8.** Section displacement and plastic strain distribution of the left line.

*4.4. Elasto-Plastic Analysis of Deformation and Stability of Nearby Surrounding Rocks and Overlying Strata of the Tunnel*

From the displacement cloud maps of the west bank levee in Figures 9–12, it can be observed that the earlier tunnel excavation results in a settlement zone near the tunnel, with the largest settlement occurring directly above the tunnel. Plastic strain zones mainly occur in the surrounding rocks and overlying soil layers near the tunnel's excavation. The latter tunnel's excavation affects the displacement distribution of the earlier tunnel, causing the deformation center to shift towards the overlying strata above the latter tunnel. Shear zones form in the soil between the settlement zone and the surrounding non-settlement zones due to the lack of displacement coordination. The plastic strain in the overlying soil layer mainly occurs in this region.

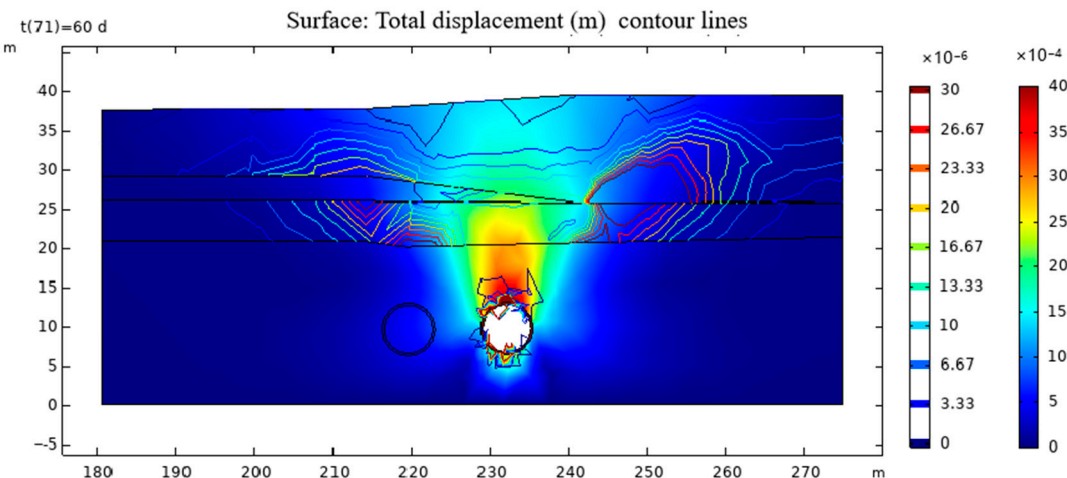

**Figure 9.** Right line shows the cross section's displacement and plastic strain distribution through the posterior eastern levee.

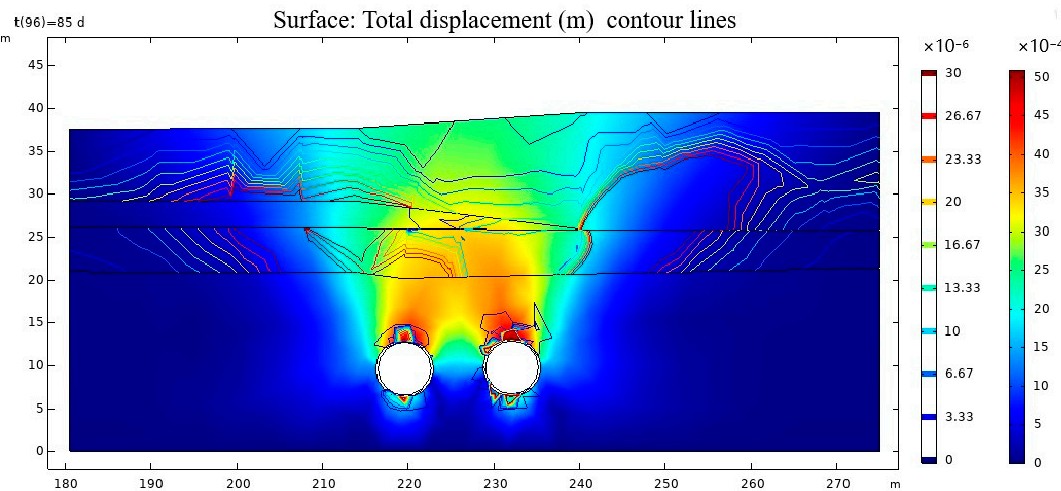

**Figure 10.** Double tunnel cross section displacement and plastic strain distribution through the rear east levee.

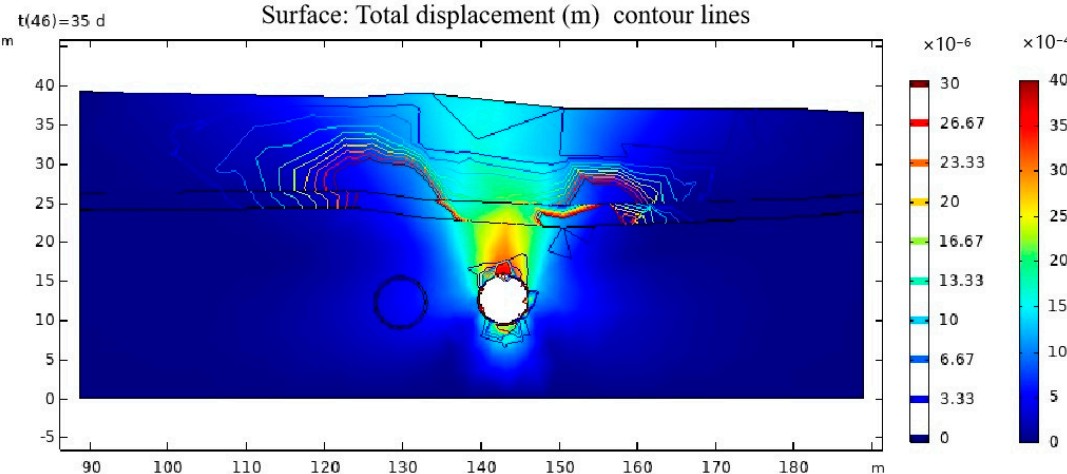

**Figure 11.** Section displacement and plastic strain distribution of the western levee after the right line.

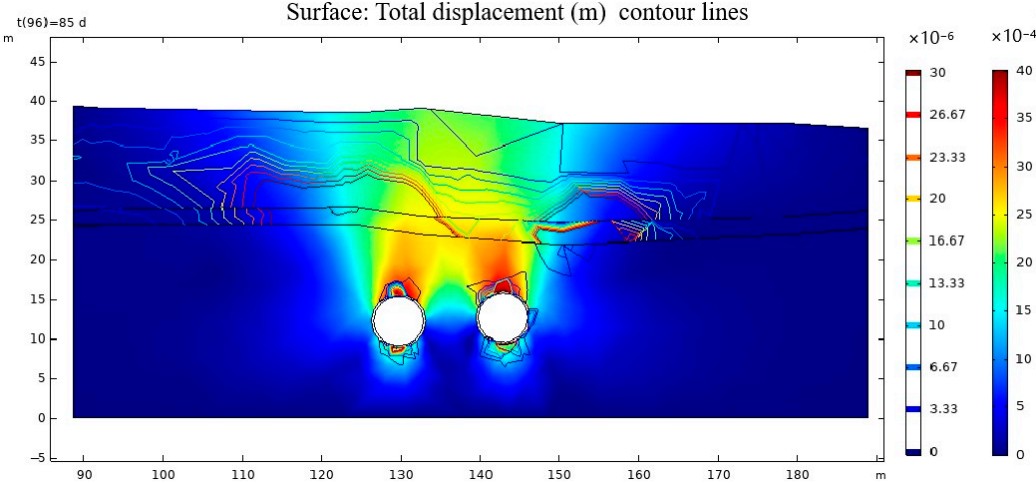

**Figure 12.** Cross section displacement and plastic strain distribution of double lines through the posterior west levee.

When the left-line tunnel is also excavated, the overlapping part of the settlement zones formed by the left-line and the right-line tunnels is located on the left side of the

settlement ellipse of the right-line tunnel, as observed from the contours of plastic strain. The expansion is based on the plastic strain zone formed by the right-line tunnel, and the settlement and shear zones are distributed as shown in Figures 13 and 14.

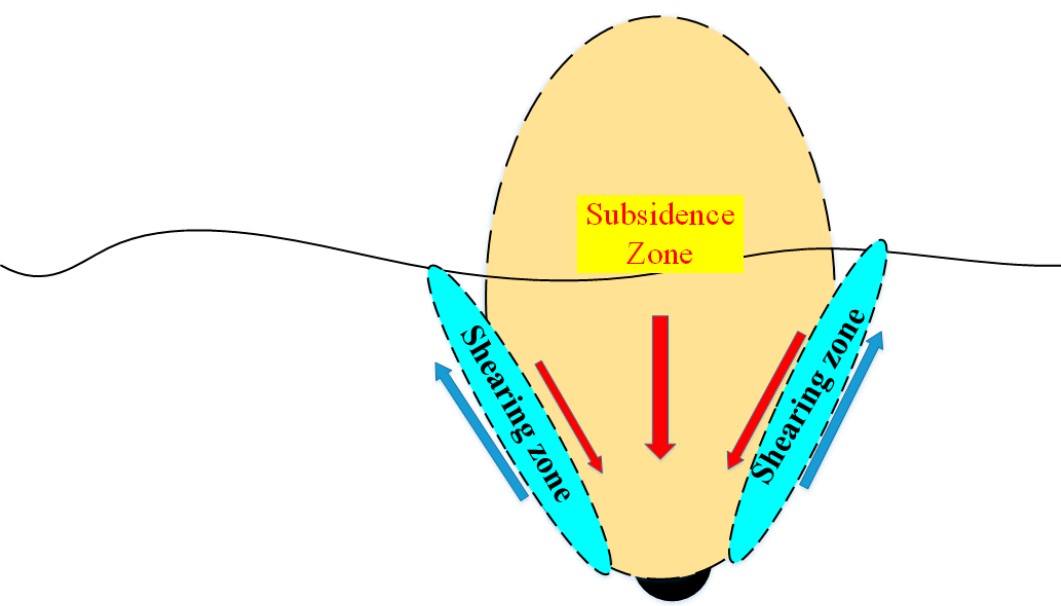

**Figure 13.** Schematic diagram of distribution of single line through posterior sedimentation and shear regions.

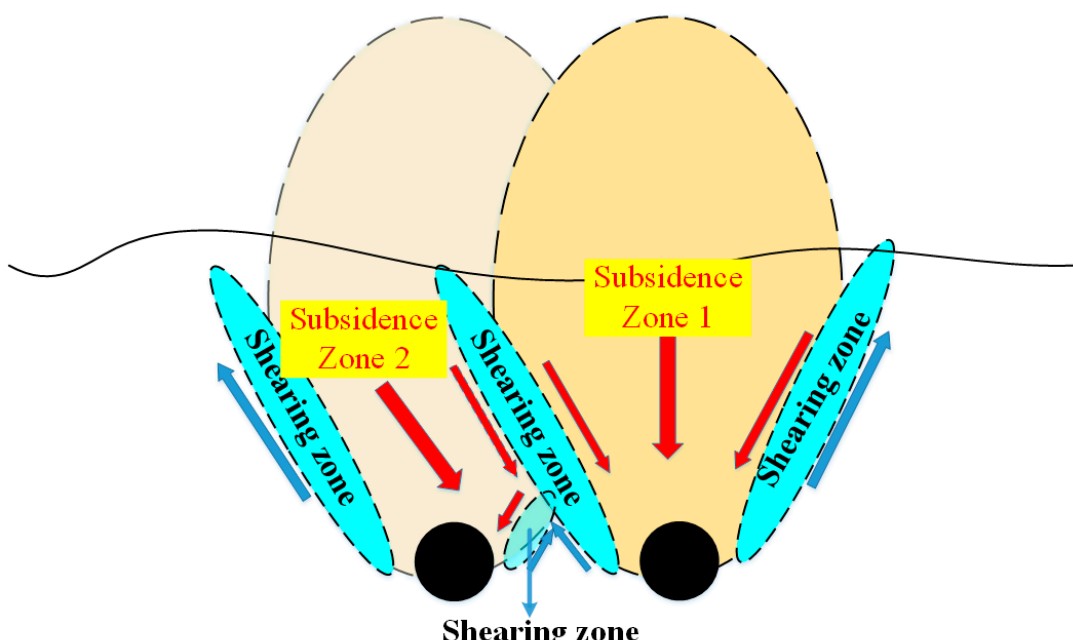

**Figure 14.** Schematic diagram of the distribution of double lines through the posterior sedimentation region and shear regions.

The elasto-plastic deformation of the surrounding rocks and overlying strata of the tunnel can be observed based on the contours of plastic strain. Plastic strains are mainly concentrated in the roof and floor of tunnel. The plastic strain zone generated by the earlier tunnel excavation is larger than that of the latter tunnel, as the surrounding rocks experience some degree of stress release after the excavation of the earlier tunnel.

### 4.5. Analysis of Surface Displacement of the Levee above the Tunnel

As shown in Figures 15 and 16, with the progress of tunnel excavation, when the length of the leading tunnel excavation reaches 320~390 m and 600~670 m, respectively, funnel-shaped settlement begins to occur near the surface of the west bank dike and the east bank dike (no obvious settlement holes appeared before), and the volume of settlement holes gradually expands as the leading tunnel approaches the dike. However, the centers of the settlement holes are all located directly above the leading tunnel. When the leading tunnel cuts through the flood control dike (leading tunnel excavation reaching 390~425 m and 670~715 m, respectively), before the excavation of the trailing tunnel enters the range that affects the flood control dike, the volumes and shapes of the settlement holes no longer change, indicating that surface settlement is temporarily stable. Afterwards, as the trailing tunnel excavation progresses closer, the centers of the settlement holes gradually move from above the leading tunnel, that is, the right tunnel towards the vicinity of the centerline between the two tunnels. Regarding the west bank dike, after the leading tunnel excavation reaches 490 m (trailing tunnel excavation reaches 390 m), there will be no further changes in the lateral deformation settlement holes on the surface of the west bank dike. At this point, the excavation of both the leading and trailing tunnels will no longer affect the lateral deformation of the dike's surface. As for the east bank dike, after the leading tunnel excavation reaches 770 m (trailing tunnel excavation reaches 670 m), there will be no further changes in the lateral deformation settlement holes on the surface of the east bank dike. At this time, the excavation of both the leading and trailing tunnels will no longer affect the lateral deformation of the dike's surface.

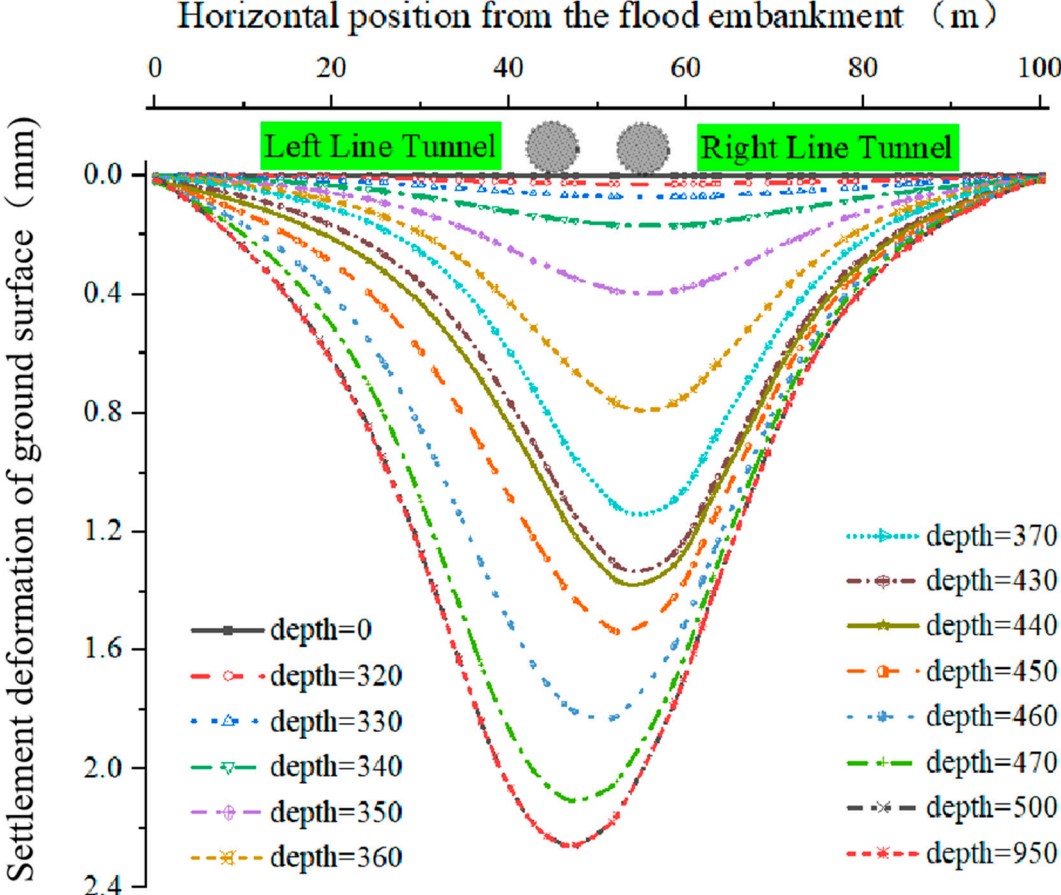

**Figure 15.** Surface displacement distribution curve of the west bank embankment at different levels of excavation progress.

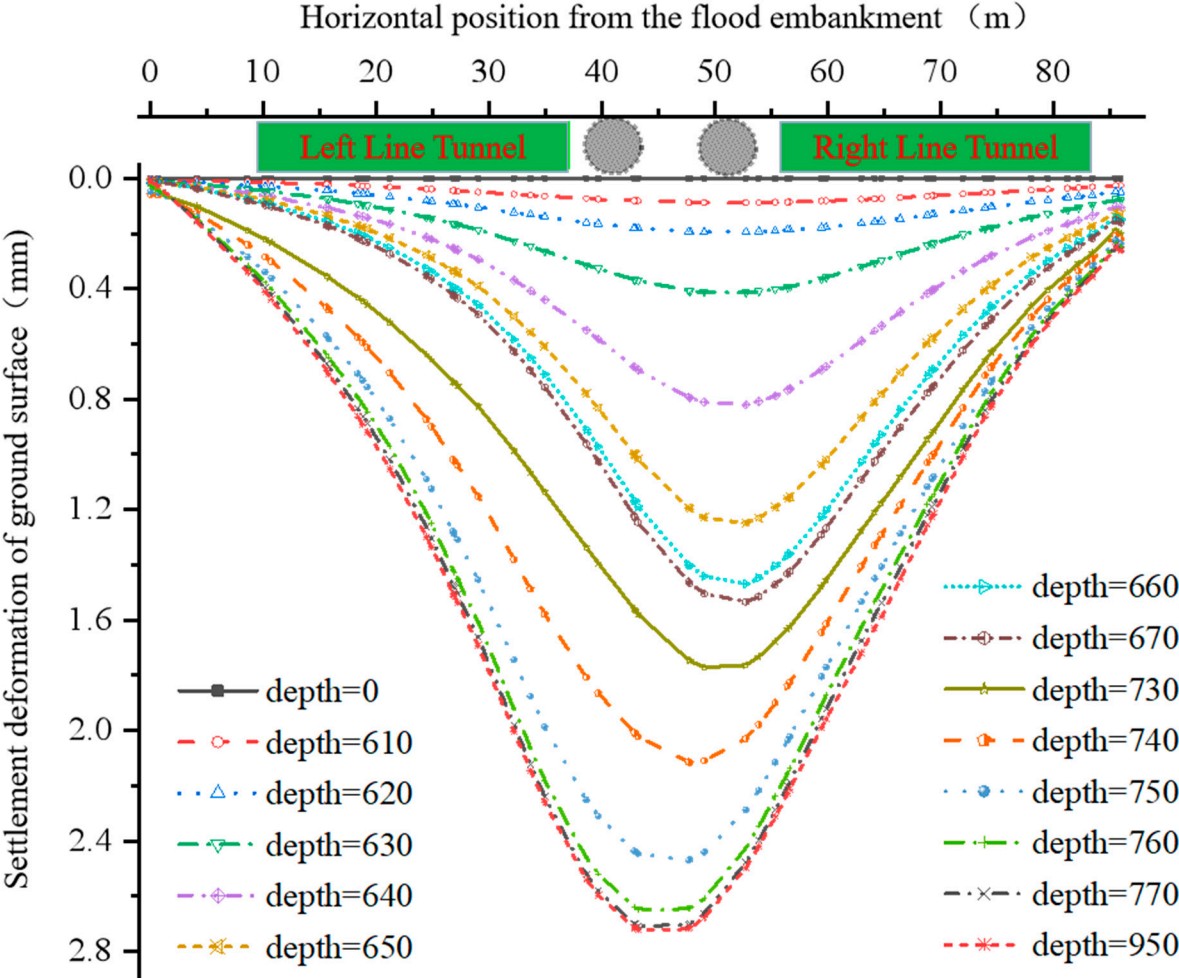

**Figure 16.** Surface displacement distribution curve of the east bank levee at different levels of excavation progress.

### 4.6. Analysis of the Effect of Excavation Progression on Surface Monitor Point Displacement

To comprehensively monitor the deformation of the embankment surface over the tunnel, four monitoring points were installed. Points 1, 2, 3, and 4 are located directly above the right-line tunnel of the west bank embankment, the centerline of the two tunnels of the east bank embankment, the right-line tunnel of the east bank embankment, and the left-line tunnel of the east bank embankment, respectively. The displacement changes over time for each monitoring point are shown in Figure 17, The location of the installation is shown in Figure 18. From the curves, the following observations can be made:

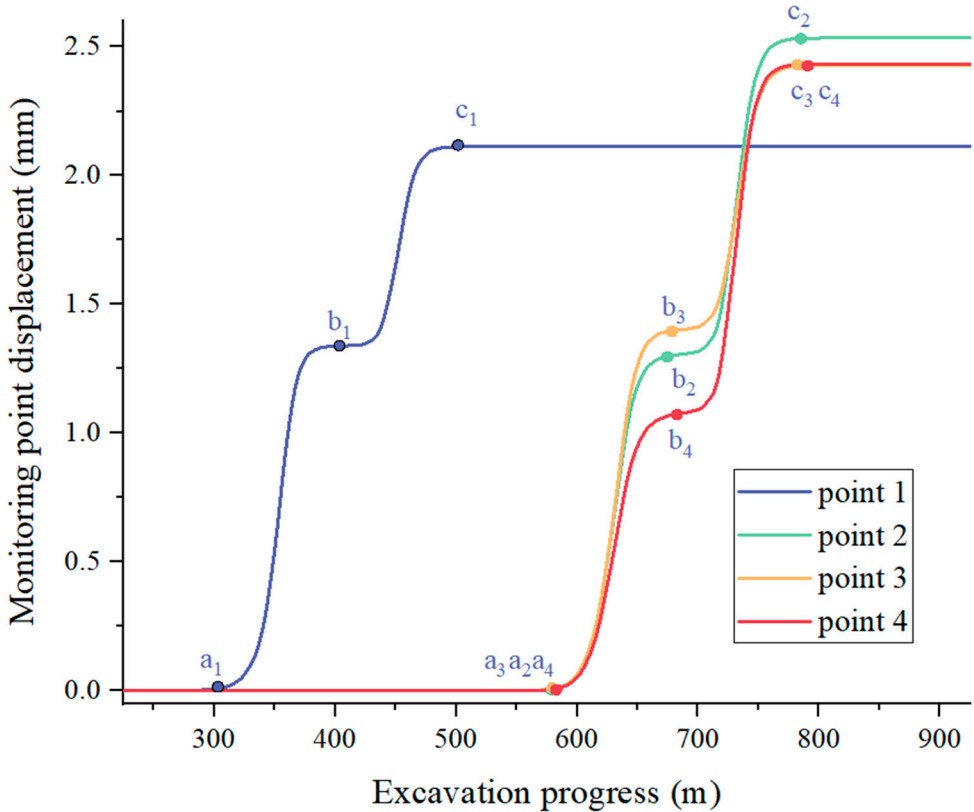

**Figure 17.** Change curve of monitoring point displacement with excavation progress.

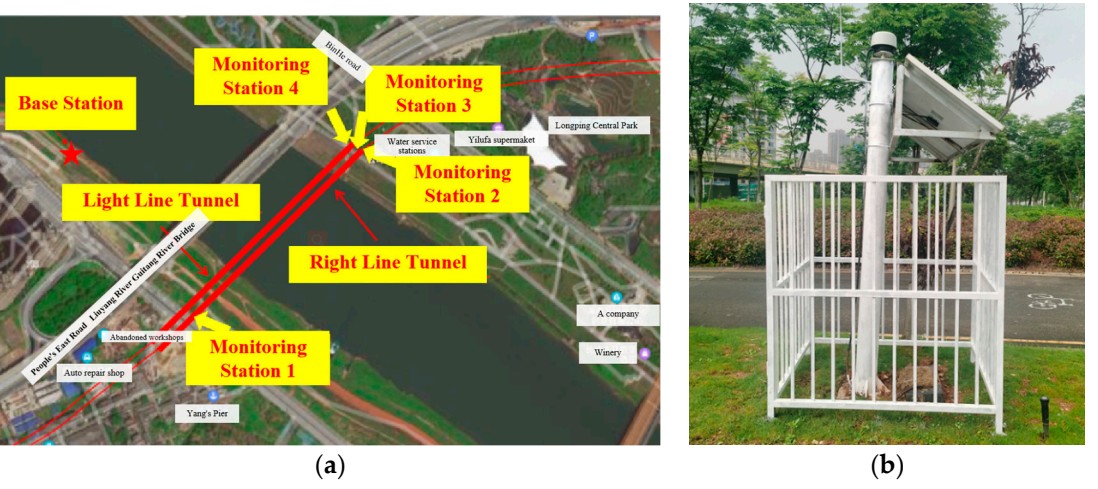

**Figure 18.** Distribution and layout of monitoring points. (**a**) Distribution Map of Monitoring Stations (**b**) Automatic monitoring station.

The surface settlement of the flood control embankment gradually sinks as the excavation progresses, and the deformation characteristics of the settlement curves at various monitoring points are similar. During the impact process of the pre-excavation tunnel, the surface settlement initially slows down, then speeds up, and finally gradually slows down again. Among them, during the rapid settlement phase, the settlement increases almost linearly until the pre-excavation tunnel reaches outside the influence zone of the tunnel settlement deformation, and the surface of the flood control embankment tends to stabilize. When the excavation of the subsequent tunnel gradually enters the settlement influence zone of the flood control embankment, the surface settlement of the flood control embankment undergoes a similar settlement process once again. In addition, it is worth

noting that the final settlement deformation of the east bank embankment is greater than that of the west bank embankment, which is consistent with the results analyzed earlier.

At monitoring point 1, there is no displacement from point 0 to point a. When the excavation of the right tunnel starts, it begins to affect monitoring point 1, causing the displacement to continuously increase. After reaching point b, the right tunnel passes through the embankment, and the displacement tends to level off. Then, due to the influence of the subsequent left tunnel, the displacement continues to increase. Since monitoring point 1 is farther from the left tunnel than from the right tunnel, the second curve is smoother than the first one, and the surface settlement deformation caused by the subsequent tunnel is smaller than that caused by the pre-excavation tunnel. After reaching point c, both tunnels have passed through the west side embankment, and there is no further change in the displacement of the monitoring point. The displacement variation process at monitoring point 2 is similar to that at monitoring point 1, and the increase in displacement caused by the passage of the two tunnels is approximately the same. Because monitoring point 2 is located at the centerline between the two tunnels, its distance from both tunnels is roughly equal, so the impact of the excavation of the two tunnels on it is similar in magnitude. In addition, the curve of monitoring point 3 is relatively shifted backward compared to monitoring point 1 because the direction of tunnel excavation is from west to east, so it passes through the west bank embankment first and then the east bank embankment. Comparatively, during the process of the right tunnel passing through the embank, the displacement slope of monitoring point 4 is large during the passage of the left line tunnel through the embankment, because the monitoring point 4 is far from the leading tunnel, the displacement of monitoring point 4 caused by the passing of the leading tunnel through the embankment is small, and the displacement of monitoring point 4 caused by the subsequent tunnel is large.

## 5. Monitoring Point Layout and Data Processing

To monitor the long-term deformation of the flood control embankment above the cross-river tunnel during construction and operation, GPS automatic monitoring stations were installed prior to the tunnel excavation. A total of four automatic monitoring points were set up as shown in Figure 18, indicating the distribution and location of the monitoring markers. Among them, monitoring point 1 is positioned directly above the right-line tunnel of the flood control embankment on the west bank, monitoring point 2 is situated in the middle of the two tunnels of the flood control embankment on the east bank, monitoring point 3 is placed directly above the right-line tunnel of the flood control embankment on the east bank, and monitoring point 4 is located directly above the left-line tunnel of the flood control embankment on the east bank. Monitoring Point 1 is taken as an example for analysis.

The presence of large-scale machinery conducting surface operations at the construction site, combined with the frequent passage of large vehicles on nearby bridges and roads, as well as the influence of wind, has caused significant fluctuations in the GPS automatic monitoring data. Some points exhibit a high concentration, while others demonstrate a considerable dispersion. In order to address this issue, the debouncing filtering algorithm was implemented to filter the raw monitoring data for four times, removing substantial noise. The remaining points were then subjected to the adjacent averaging method for further smoothing. The monitoring curve of monitoring station 1 after processing is presented in Figure 19. Among them, a, b, c, d, and e respectively indicate the monitoring time corresponding to different stages, and the purple dots correspond to the end time of different stages.

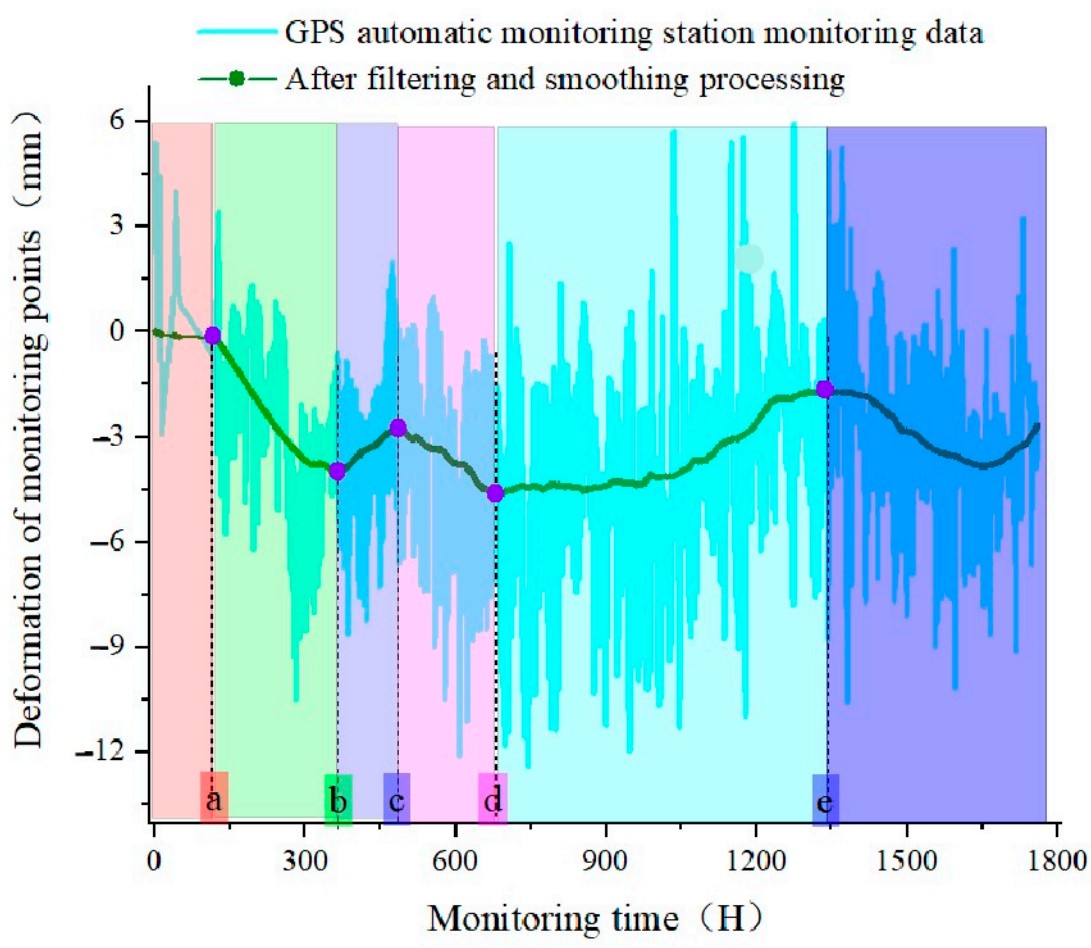

**Figure 19.** Monitoring curves filtered by the adjacent averaging method.

The results indicate that the displacement of the monitoring point remains generally stable within the range of ±5 mm. The displacement observed in sections a, b, and c is attributed to the self-settlement of the concrete foundation of the monitoring point, while the displacement in section c and d corresponds to the movements of the monitoring point when the shield cutter passes through the embankment, which aligns closely with the simulated settlement value. Even after the shield cutter completes its passage, the monitoring point still exhibits significant displacement fluctuations. It is suggested that this phenomenon may be attributed to the delay in shield support and grouting. Shield support and grouting exert pressure on the tunnel wall, leading to surface displacement fluctuations. Nonetheless, the displacement of the monitoring points remains generally stable within a range of 5 mm, under the requirement of minimal surface displacement. As a result, the stability of the flood control embankment can be ensured.

## 6. Conclusions

(1) During the process of traversing the embankment, the overlying soil layer and top and bottom plates of the tunnel undergo plastic deformation at a small scale. Notably, the deformation observed when crossing the embankment on the east bank exceeds that on the west bank, rendering the crossing on the east bank a relatively riskier stage of the overall project;

(2) During tunnel excavation, due to the displacement mismatch between the settlement area and the surrounding unconsolidated area, a shear zone emerges, with the majority of plastic deformation in the overlying soil layer occurring within this region. The shear zone that forms subsequent to the excavation of the subsequent tunnel primarily appears on the left side of the settlement ellipse. The plastic deformation zone

originating from the preceding tunnel expands further, with the plastic strain zone generated by the near-field surrounding rock of the preceding tunnel exceeding that of the subsequent tunnel;

(3) An analysis of the displacement field and plastic strain distribution of the embankment section enables the identification of areas in which the plastic strain may arise within the embankment under the present construction conditions. The tunnel excavation process affects the final plastic zone of the overlying rock layer. As the excavation progresses, a funnel-shaped settlement displacement gradually emerges during the excavation of the preceding tunnel. Subsequently, this settlement funnel gradually widens during the excavation of the subsequent tunnel. The point of maximum settlement gradually transitions from directly above the preceding tunnel to the middle of the two tunnels;

(4) The debouncing filtering algorithm and adjacent averaging method were employed to filter and smoothen the GPS automatic displacement monitoring data obtained from the on-site installations, respectively. The results indicate that the deformation of the monitoring points remains within the range of 5 mm, with the final monitoring deformation results being consistent with the numerical simulation outcomes. As such, the tunnel excavation exerts a minimal impact on the surface displacement of the flood control embankment.

**Author Contributions:** Conceptualization, Q.Z. and J.L.; Methodology, Y.L. and P.C.; Formal analysis, Q.Z. and J.L.; Investigation, Q.Z. and J.L.; Resources, P.C. and K.L.; Data curation, Y.L. and J.L.; Writing—original draft, Q.Z. and Y.L.; Writing—review and editing, K.L. and Y.P.; Supervision, K.L.; Funding acquisition, P.C. and K.L. All authors have read and agreed to the published version of the manuscript.

**Funding:** This research was funded by (1) the Water Conservancy Science and Technology Major Project of Hunan Province, China (No. XSKJ2019081-10), (2) the National Natural Science Foundation of China (No. 52204119), (3) the Natural Science Foundation of Hunan Province, China (No. 2023JJ40729), and (4) the Fundamental Research Funds for the Central University of Central South University (No. 2019zzts667).

**Institutional Review Board Statement:** Not applicable.

**Informed Consent Statement:** Not applicable.

**Data Availability Statement:** Not applicable.

**Acknowledgments:** This work was financially supported by the Water Conservancy Science and Technology Major Project of Hunan Province, China (No. XSKJ2019081-10), the National Natural Science Foundation of China (No. 52204119), the Natural Science Foundation of Hunan Province, China (No. 2023JJ40729), and the Fundamental Research Funds for the Central University of Central South University (No. 2019zzts667). The authors would also like to acknowledge the editors and reviewers for their constructive comments, which have greatly improved this paper.

**Conflicts of Interest:** The authors declare no conflict of interest.

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
