# Peer review of "A Study on the Influence of Double Tunnel Excavations on the Settlement Deformation of Flood Control Dikes"

_sustainability, doi:10.3390/su151612461_

Round 1
Reviewer 1 Report
This article presents a nice study on the influences of double tunnel excavation on the settlement deformation of flood control dikes by numerical simulation and field monitoring methods. It establishes a complete model for the excavation of subway section tunnels and develops a new excavation condition function. Topic falls into the scope of the journal and possesses strong innovative points. However, certain modifications on the language and reporting style are required before the publication.
1.The main focus of the article is on the use of the software 'COMSOL Multiphysics, but its application in tunnel excavation is not mentioned in the literature review.
2. Please label the names of the two images in Fig. 18 separately.
3. The conclusion (1) contradicts the results analyzed earlier in the article. Please carefully review it.
4. Some points in Figs. 5 to 10 needs to be modified according to the requirements of the image format.
5. The main purpose of this article is to study the stability of the overlying flood control embankment during the excavation of river-crossing tunnels. The key terms should include "river-crossing tunnel," and it is suggested to replace the term "excavation process" with "river-crossing tunnel."
6. Why is the debounce filtering algorithm adopted for data filtering in the process of monitoring data handling?
7. Why is the range of the abscissa in Fig. 4 from 97500 to 98300 meters?
Author Response
This article presents a nice study on the influences of double tunnel excavation on the settlement deformation of flood control dikes by numerical simulation and field monitoring methods. It establishes a complete model for the excavation of subway section tunnels and develops a new excavation condition function. Topic falls into the scope of the journal and possesses strong innovative points. However, certain modifications on the language and reporting style are required before the publication.
Point 1: The main focus of the article is on the use of the software 'COMSOL Multiphysics, but its application in tunnel excavation is not mentioned in the literature review.
Response 1: According to your suggestions, we have added references to the previous achievements in numerical simulation in the article, focusing on summarizing the results of COMSOL in the study of the impact of tunnel excavation on surface settlement and deformation. The content is as follows:
Deng et al [7] developed a three-dimensional finite element numerical model in a compo-site stratum using the software "COMSOL". Through the utilization of fluid-structure coupling and multi-physics field analysis, the study derived the laws governing ground settlement deformation and examined the impact of four factors, e.g., tunnel face spacing, tunnel spacing, tunnel depth and water level on the riverbed, on the ground settlement curve of double-hole tunnels. Ying et al. [28] used COMSOL to establish a numerical model and carried out parameter analysis on the seabed thickness, permeability coefficient, tunnel buried depth and lining thickness involved in the construction of subsea tunnels. COMSOL Muti-physics simulation results showed that the error caused by the seabed thickness could not be ignored in tunnel construction.
Point 2: Please label the names of the two images in Fig. 18 separately.
Response 2: We have provided corresponding explanations below Fig. 18 in the text. The content of the explanation is as follows: (a) Distribution Map of Monitoring Stations (b) Automatic monitoring station.
Point 3: The conclusion (1) contradicts the results analyzed earlier in the article. Please carefully review it.
Response 3: Thanks for your comments. We have made modifications according to your advice. The most dangerous area should be on the east bank, rather than the west bank.
Point 4: Some points in Figs. 7 to 10 needs to be modified according to the requirements of the image format.
Response 4:According to your suggestions, we have thoroughly examined the issues in the images. We have translated the Chinese text in figures 7 to 10 into English and replaced the corresponding images accordingly.
Point 5: The main purpose of this article is to study the stability of the overlying flood control embankment during the excavation of river-crossing tunnels. The key terms should include "river-crossing tunnel," and it is suggested to replace the term "excavation process" with "river-crossing tunnel."
Response 5:Thanks for your comments. We have already made the modifications according to your suggestions.
Point 6: Why is the debounce filtering algorithm adopted for data filtering in the process of monitoring data handling?
Response 6: Due to the surface operations of large-scale mechanical equipment at the construction site, as well as the passing of large vehicles on the surrounding bridges and roads, and the influence of wind force, the GPS monitoring data fluctuates greatly. In the three-dimensional space, it can be observed that some points are highly concentrated, while others have a high degree of dispersion.
How to eliminate the fluctuations caused by these accidental factors? In signal processing, commonly used filtering methods include the arithmetic mean filtering, amplitude limit filtering, first-order lag filtering, recursive average filtering, debouncing filtering, median filtering, median mean filtering, IRR digital filter, amplitude mean filtering, weighted recursive average filtering, and amplitude debouncing filtering. However, from the perspective of monitoring raw data, it is not appropriate to simply use a certain arithmetic mean filtering method for curve smoothing, because some fluctuation values have far exceeded the effective displacement value. These fluctuations need to be removed first and then recovered after a certain period of time. This displacement can be regarded as "elastic vibration", which needs to be completely eliminated. Among the aforementioned filtering methods, debouncing filtering can achieve this purpose. The principle of the debouncing filter is to set a threshold for vibration and a counter. The difference between each sampling value and the current valid value is calculated. If the difference is less than the threshold, the counter is set to zero; if the difference is greater than the threshold, the counter is increased by 1, and the counter value is checked to see if it exceeds the set upper limit. If the counter overflows, the current sampling value is replaced with the current valid value, and then the counter is cleared. Through this filtering method, the original monitoring data can remove significant noise and obtain the following waveform.
Additionally, it can be found that the amplitude of the fluctuations has significantly decreased. However, this filtering method can only process data in one direction, and the filtered data still have significant fluctuations. Another method can be used to improve the accuracy of the results. We know that if we do not consider the actual displacement, the displacement of the monitoring point is pure noise and should follow a normal distribution. From the perspective of displacement in the x and y directions, this is indeed the case. The change in our monitoring data is smaller than the amplitude of the noise. However, we can replace the monitoring values with the average values over a certain period of time. In three-dimensional space, the average value of coordinates is the position where all monitoring data are relatively concentrated. When the distance between a monitoring data point and the average value point is greater than 10mm, it indicates that this point deviates significantly from the actual position of the monitoring point, and we discard such points.
Point 7: Why is the range of the abscissa in Fig. 4 from 97500 to 98300 meters?
Response 7: X value represents the coordinate value of the tunnel in the east-west direction in the model and corresponds to the X coordinate value of the model.
Reviewer 2 Report
The authors presented a well-written manuscript about a study regarding the influence of double tunnel excavation on the settlement deformation of flood control dikes combining numerical simulation and field monitoring methods. One innovation of the manuscript is to establish a computational model of the entire subway tunnel section crossing a flood control embankment under the action of fluid-structure interaction by means of AutoCAD, 3Dmine, and COMSOL Multiphysics. Topic falls into the scope of the journal and the conclusions provide useful new information on the topic, although I can spot several trivial mistakes. I recommend publication after the following modifications have been made.
Specific comments:
1. The labeling of the vertical axis in Fig. 15 "Lateral settlement deformation of ground surface (mm)" is unclear and should be changed to "settlement deformation of ground surface (mm)."
2. In conclusion 1), although deformation is greater crossing the east bank dike than the west bank dike, it is said that crossing the west bank dike is a relatively high-risk stage of the entire project.
3. Translate the Chinese text in Figs. 5 and 6 into English.
4. Please specify the meanings of the X and Y axes in Figure 4.
5. Why is the deformation near the surface area smaller in Figs. 7 and 8 compared to the deformation near the tunnel area?
6. Please provide a literature review on the application of the software "COMSOL Multiphysics" in the tunnel excavation.
7. One should emphasize the particularity of the geological conditions at this site and the significance of the research in engineering geology.
8. Translate the Chinese text in Fig. 18 into English.
Some minor grammatical errors need to be improved
Author Response
The authors presented a well-written manuscript about a study regarding the influence of double tunnel excavation on the settlement deformation of flood control dikes combining numerical simulation and field monitoring methods. One innovation of the manuscript is to establish a computational model of the entire subway tunnel section crossing a flood control embankment under the action of fluid-structure interaction by means of AutoCAD, 3Dmine, and COMSOL Multiphysics. Topic falls into the scope of the journal and the conclusions provide useful new information on the topic, although I can spot several trivial mistakes. I recommend publication after the following modifications have been made.
Specific comments:
Point 1: The labeling of the vertical axis in Fig. 15 "Lateral settlement deformation of ground surface (mm)" is unclear and should be changed to "settlement deformation of ground surface (mm)."
Response 1: The modifications have been made to Figure 15 according to your suggestions.
Point 2: In conclusion 1), although deformation is greater crossing the east bank dike than the west bank dike, it is said that crossing the west bank dike is a relatively high-risk stage of the entire project.
Response 2: We have carefully considered the suggestions you provided and find your opinions very valuable. We have made modifications according to your advice. The most dangerous area should be on the east bank, rather than the west bank.
Point 3: Translate the Chinese text in Figs. 7 and 8 into English.
Response 3: The modifications have been made to Figs. 7 and 8 according to your suggestions.
Point 4: Please specify the meanings of the X and Y axes in Figure 4.
Response 4: We have provided corresponding explanations below Figure 4 in the text. The explanation for this is as follows: X value represents the coordinate value of the tunnel in the east-west direction in the model and corresponds to the X coordinate value of the model. Y represents the coordinate value of the tunnel in the north-south direction in the model.
Point 5: Why is the deformation near the surface area smaller in Figs. 7 and 8 compared to the deformation near the tunnel area?
Response 5: When the tunnel depth is lower, the arching effect of the soil can mitigate the surface settlement, leading to smaller deformations.
Point 6: Please provide a literature review on the application of the software "COMSOL Multiphysics" in the tunnel excavation.
Response 6: According to your suggestions, we have added references to the previous achievements in numerical simulation in the article, focusing on summarizing the results of COMSOL in the study of the impact of tunnel excavation on surface settlement and deformation. The content is as follows:
Deng et al [7] developed a three-dimensional finite element numerical model in a composite stratum using the software "COMSOL". Through the utilization of fluid-structure coupling and multi-physics field analysis, the study derived the laws governing ground settlement deformation and examined the impact of four factors, e.g., tunnel face spacing, tunnel spacing, tunnel depth and water level on the riverbed, on the ground settlement curve of double-hole tunnels. Ying et al. [28] used COMSOL to establish a numerical model and carried out parameter analysis on the seabed thickness, permeability coefficient, tunnel buried depth and lining thickness involved in the con-struction of subsea tunnels. COMSOL Multi-physics simulation results showed that the error caused by the seabed thickness could not be ignored in tunnel construction.
Point 7: One should emphasize the particularity of the geological conditions at this site and the significance of the research in engineering geology.
Response 7: We have already provided explanatory content in the section "2.1 Engineering Geological Conditions" in the paper. The contents of the explanation are as follows: Therefore, when water infiltrates the rock layers, it weakens the surrounding rock, which poses a substantial risk to the stability of tunnel.
Point 8: Translate the Chinese text in Fig. 18 into English.
Response 8: The modifications have been made to Fig. 18 according to your suggestions.
Point 9: Some minor grammatical errors need to be improved.
Response 9: Thanks for your comments. The language has been polished.
Round 2
Reviewer 2 Report
The author revised the paper as requested, and I believe it is ready for publication.
Minor editing of English language required